# Stable Isotope Analysis Reveals Common Teal (*Anas crecca*) Molting Sites in Western Siberia: Implications for Avian Influenza Virus Spread

**DOI:** 10.3390/microorganisms12020357

**Published:** 2024-02-09

**Authors:** Alexey V. Druzyaka, Olga R. Druzyaka, Kirill A. Sharshov, Nikita Kasianov, Nikita Dubovitskiy, Anastasiya A. Derko, Ivan G. Frolov, Jyrki Torniainen, Wen Wang, Mariya A. Minina, Alexander M. Shestopalov

**Affiliations:** 1Institute of Systematic and Ecology of Animals, Frunze Str. 11, 630091 Novosibirsk, Russia; abdrashitova-olga@mail.ru (O.R.D.); frolov.ivg@gmail.com (I.G.F.); maff14@yandex.ru (M.A.M.); 2Department of Natural Sciences, Novosibirsk State University, Pirogova Str. 2, 630090 Novosibirsk, Russia; 3Federal Research Center of Fundamental and Translational Medicine, Timakova Str. 2, 630060 Novosibirsk, Russia; sharshov@yandex.ru (K.A.S.); nikitadubovitskiy@gmail.com (N.D.); a.derko19@gmail.com (A.A.D.); shestopalov2@mail.ru (A.M.S.); 4Institute of Zoology, Ministry of Science and Higher Education of the Republic of Kazakhstan, Al-Farabi Ave. 93, Almaty 050060, Kazakhstan; 5Department of Biological and Environmental Science, University of Jyväskylä, P.O. Box 35, FI-40014 Jyväskylä, Finland; 6Open Science Centre, University of Jyväskylä, P.O. Box 35, FI-40014 Jyväskylä, Finland; 7State Key Laboratory of Plateau Ecology and Agriculture, Qinghai University, Xining 810016, China; 007cell@163.com

**Keywords:** avian influenza, migration routes, molting grounds, stable isotopes, isoscape, *Anas crecca*, common teal

## Abstract

The wetlands of southwestern Siberia (SWS) are a crossroads of bird migration routes, bringing avian influenza (AIV) strains that were previously isolated in different regions of the continent to Siberia. It is known that Anseriformes that breed in SWS migrate for the winter to central Hindustan or further west, while their migration routes to southeast Asia (SEA) remain unconfirmed. Here, we mapped the molting sites of the migrating Common Teals (*Anas crecca*) via analyzing stable hydrogen isotope content in feathers of hunters’ prey and supplemented the analysis with the genetic structure of viruses isolated from teals in the same region. Post-breeding molt of autumn teals most likely occurred within the study region, whereas probable pre-breeding molting grounds of spring teals were in the south of Hindustan. This link was supported by viral phylogenetic analysis, which showed a close relationship between SWS isolates and viruses from south and southeast Asia. Most viral segments have the highest genetic similarity and the closest phylogenetic relationships with viruses from teal wintering areas in southeast Asian countries, including India and Korea. We assume that the winter molt of SWS breeding teals on the Hindustan coast suggests contacts with the local avifauna, including species migrating along the coast to SEA. Perhaps this is one of the vectors of AIV transmission within Eurasia.

## 1. Introduction

The effective control of the spread and prevention of zoonotic infections, such as Avian influenza, requires a comprehensive understanding of the structure of the transmission pathways and the transmission of pathogens between natural hosts and from them to humans. It has been established that more than 60% of global infectious diseases are of zoonotic origin, and among those that have emerged over the past 40 years, zoonotic infections comprise more than 75% [1]. Drivers of zoonotic transmission include climate and weather, economic development, poverty and social inequality, war and famine, human–wildlife interactions, and land use and ecosystem changes. Anthropogenic pressure on the environment, the mobility of the population and the emergence of new points of contact between economic activity and wildlife lead to an increase in the relative role of natural zoonoses. One of the current threats is influenza A virus (AIV) circulating in wild and domestic bird populations [2]. The natural reservoir for the influenza virus is waterfowl, which can serve as a carrier and spreader of the virus [3]. In addition, it is the waterfowl that are considered the most likely candidate for the role of the main distributor of highly pathogenic avian influenza from endemic pestholes in southeast Asia to the Eurasian continent [4,5,6] and even to North America [7].

The southern part of western Siberia is located at the intersection of three transcontinental flyways [8,9] and at the same time, being rich in wetlands, is an area of nesting and migration stopover for more than 50 species of Anseriformes [10]. Previously, a phylogenetic relationship was established between low pathogenic strains of influenza A virus isolated from ducks during migration in western Siberia and strains circulating in different parts of Eurasia, such as the north of western Europe and southeast Asia [11,12,13]. In addition, the highly pathogenic AIV strains that caused the mass death of poultry in 2005 in the south of western Siberia turned out to be phylogenetically similar to the strains that caused the mass death of birds on Qinghai Lake in northern China in the same year [14,15]. This indicates the presence of diversified transmission routes for type A influenza viruses via migratory birds within the Eurasian continent that cross in the south of western Siberia. To understand the mechanisms of this and to control the transfer of dangerous virus strains across Eurasia, it is crucial to know the routes and timing of waterfowl migrations through western Siberia, including the possibility of their direct or indirect contact with the avifauna and domestic birds of southeast Asia.

Traditionally, migratory flows of birds have been studied by analyzing large volumes of ring recovery data. However, the numerous limitations of this classical method are well known today. Mainly, these are a low rate of recoveries relative to the number of ringed birds and, hence, comparatively slow accumulation of material. In addition, the analysis of ring recoveries provides insight into only a few (usually two) points of the migration path of each bird, which are, moreover, chosen subjectively by the ringer and the presence of humans. As a result, the impact of, for example, climate change on migration routes or the transformation of the subspecies/population structure of a species may elude the attention of ringers [16,17,18]. Combining the classical approach with modern methods, such as tracking several individuals using various types of transmitters [19] and the analysis of stable isotope (SIA) content in feathers [20], makes it possible to clarify various issues in the biology of migration that are inaccessible with the traditional methods only [21]. In particular, the stable isotope method makes it possible to map the subspecies and population structures of migrant species and to clarify migration relationships between different geographical populations (see the review in [22]). In addition, stable isotope analysis (SIA) provides qualitatively new opportunities for understanding the transmission of carry-over effects such as zoonoses, since both healthy and infected individuals can be included in the analysis, which makes it possible to compare their places of origin and flight routes. In contrast, at the moment of marking birds with transmitters or rings, there is no way to know if an individual is infected. Therefore, there is no way to distinguish whether pathogens’ carriers’ migration patterns differ from those of other populations. Also, the number of transmitters available to the researcher is usually limited. This has possibly led to misinterpretations in previous attempts to reveal migration links between pestholes of avian influenza via bird tracking [23]. We believe that the SIA method gives a good perspective for understanding the transmission of avian influenza in Eurasia, although at the moment there are relatively few studies in this field [24,25]. A recently published monograph relates to this field [26] (but see [27]), is devoted to the prospect of mapping the migration routes of direct AIV carriers using stable isotope analysis. It seems very likely that there will be a further increase in research in this area in Eurasia.

In western Siberia, waterfowl migrations were studied in detail in the 1970s and 1980s by numerous ringing experiments. The mapping of ring recoveries has clarified the main wintering areas for waterfowl breeding in the southern part of western Siberia. These were vast regions that include Hindustan, the south of the Caspian Sea, the eastern Mediterranean, and even Northern Europe [28]. However, neither this nor any other literary source provide any recovery data from Southeast Asia, just with rare exceptions (single ring recoveries from Japan from the following species: Common Pochard *Aythya ferina*, Common Pintail *Anas acuta*, Wigeon *Anas penelope*, see [28]). At the same time, it was shown in the same years that there was an intensive migration of various bird species, including waterfowl, between the southeastern tip of the West Siberian Plain and the Dzungarian depression through the so called Dzungarian Gates [29,30]. Thus, direct migratory flights of waterfowl nesting in western Siberia to southeast Asia travel through the aforementioned Dzhungarian Gates along the chain of lakes through Sasykkol–Alakol–Jalanashkol–Ebinur–Bagrashkol–Lobnor and further in a southeasterly direction to Qinghai Lake or along the Ili River valley in the same direction. With the help of GPS tracking, the last route was confirmed for Bewick’s Swan (*Cygnus bewickii*) from the nesting site in the lower reaches of the Ob River to the beginning of the Ili River [31]. In addition, no influenza viruses were isolated in western Siberia, and influenza outbreaks among poultry were never recorded north of the forest steppe subzone. The reasons for this are not entirely clear, taking into account that Anatids used to migrate through the influenza-positive south of the region to their nesting sites in its northern part [28]. At the same time, the SIA approach clearly revealed the migration of some Anseriformes populations nesting in the north of western Siberia to western and central Europe for wintering, such as the Bean Goose (*Anser fabalis*) [19] and Common Teal [18]. Obviously, a further understanding of the ways and mechanisms of transmission of the avian influenza virus to Europe from southeast Asia requires clarification of the current state of flyways and the population structure of influenza carrier species nesting in western Siberia. Practical investigations in this field are especially important as actual data are lacking, because in the last 30 years, the ringing of waterfowl in this region has practically ceased, and neither tracking, with the exception of the mentioned work by Vangeluwe et al. [31], nor the analysis of stable isotopes in feathers have been used either.

In this work, we applied the SIA methods to feathers to clarify the current state of the migration routes of one of the most common species of waterfowl nesting and flying through the south of western Siberia—the Common Teal. The species was chosen due to its high frequency of transmission of type A influenza viruses, its vast and diverse nesting area, as well as a wide range of migratory directions [32]. Isotope analysis was used to determine the places of post-breeding and pre-breeding molting of birds that migrated in autumn 2017 and in spring 2018 through the study area, respectively.

To determine Common Teal molting regions, we chose hydrogen stable isotope analysis, since its composition in feathers (δ^2^H_f_) better reflects the same in precipitation hydrogen (δ^2^H_p_) via waterfowl diet in molting or hatching areas [33,34]. Very stagnated and predictable global spatiotemporal patterns of δ^2^H_p_ are summarized in the form of the so-called isotope landscapes, or isoscapes [35,36]. This approach is widely used to solve various environmental issues, including mapping animal migrations (see [36] for a review). Deuterium in precipitation tends to deplete from the equator toward the poles and simultaneously inland from the coasts of continents. For northern Eurasia, these trends are expressed mainly in the SWS–ENE direction, which, in turn, makes it possible to map the seasonal migrations of birds, recording their molting locations [33]. In this work, we used δ^2^H_f_ analysis to locate the current molting sites for Common Teals migrating through the south of western Siberia, to compare with such locations mapped earlier. We tried to develop ideas about how we could connect the data on stable isotope analysis of the migrating teals through transcontinental flyways with the routes of avian influenza transmission.

## 2. Materials and Methods

### 2.1. Ethical Issues and Sample Collection

The present study was conducted in accordance with the approval and requirements of the Biomedical Ethics Committee of the Federal Research Center of Fundamental and Translational Medicine (FRC FTM), Novosibirsk (Protocol Nos. 2013-23 and 2021-10). The bird specimens were collected during the state hunting season with a license from the regional Ministries of Ecology and Natural Resources as part of the annual collection of biological material (the Programme for the Study of Infectious Diseases of Wild Animals, FRC FTM, Novosibirsk). Biological material was collected from autumn 2017 to autumn 2018 from the Common Teal hunted during migration. Teals were sampled in two regions of the south of western Siberia: Novosibirsk (53.60° N and 77.60° E, sampling area nearly 50 × 30 km, autumn 2017 and spring and autumn 2018) and Tomsk region (57.60° N and 83.90° E, collection area about 15 × 20 km, only spring 2018).

For the isotope analysis, from each autumn migrant, the 7th and 8th primary flight feathers were collected, for a total of 18 samples; from spring migrants, we chose plumage parts, which tended to be the last to grow during the pre-breeding molting, for a total of 10 spring samples. In males, we collected scapulars, and in females, these included nape and uppertail coverts. These plumage parts were already replaced in winter during pre-breeding molting in some species of dabbling ducks [37]. Sampling was conducted from 2 to 12 September in 2017 and from 20 to 30 April in 2018.

Sampling for avian influenza was carried out as part of a number of long-term monitoring studies on avian influenza virus in Siberia. During avian influenza monitoring in 2018, a sampling of 50 Common Teals in spring season and 73 Common Teals preyed by amateur hunters was carried out. Species, sex, and age of birds were registered (only for autumn migrants, juveniles were distinguished from individuals over 1 year old). Sampling to detect avian influenza virus was carried out according to WHO recommendations from 2006. Cloacal swabs of wild waterfowl were collected during the hunting season in individual 2 mL tubes containing 1 mL of viral transport medium. The tubes containing sample biomaterial were stored in liquid nitrogen immediately and transported to the laboratory for analysis [38].

### 2.2. Stable Isotope Analysis in Feathers

All samples were placed in an individual paper envelope and stored at room temperature until isotope analysis was performed. Analysis of the deuterium content of feathers was performed in the Isotope Laboratory of the University of Jyväskylä, Finland. Prior to analysis, feather samples were cleaned of surface oils and waxes via washing in a 2:1 (*v*/*v*) solution of chloroform/methanol for 24 h and dried at room temperature under a fuming hood for 48 h. The distal parts of the feather vanes were cut using a scalpel and forceps from about 1 cm from the tip to the feather basal part of the primary feather. Samples were weighed, following standard procedure, to 0.350 mg ± 0.05 mg. Between every feather, sampling instruments were cleaned with ethanol and air dried. Cut tips were placed into silver capsules, after which they were left for at least four days in a laboratory atmosphere, according to Wassenaar and Hobson guide [39]. δ^2^H_f_ was measured using an Isoprime 100 CF-SIRMS (Isoprime, Cheadle, UK) in combination with an Elementar Pyrocube analyzer (Elementar, Hanau, Germany). The results of the analysis were standardized against two reference laboratory keratin materials obtained from Environment Canada (KHS: 2H = −54.1‰ and CBS: 2H = −197‰) following from Wassenaar and Hobson [40]. The results are given in the standard δ^2^H notation as ppm (‰) difference from the international VSMOW-SLAP (International Atomic Energy Agency) standard. According to the standard deviation of repeated measurements of standard samples, the instrumental error was estimated at SD = 2.4‰. All sample measurements were numerically corrected according to the protocol of [41].

### 2.3. Statistical Analysis of Differences in δ^2^H_f_ in Teals of Different Sexes and Ages

Due to the relatively small sample size, we used the nonparametric Mann–Whitney U-test to compare δ^2^H_f_ of adult birds with juveniles and males with females in spring and autumn separately. A *p*-value of 0.05 was considered as a significant threshold.

### 2.4. Geographic Assignment of Feather Samples to Origin

The relation between teal feathers’ δ^2^H_f_ and predicted precipitation δ^2^H_p_ was estimated according to Clark et al. [42], who reported results for the Lesser Scaup, using the following formula:$$\delta^{2}H_{f}=-31.6+O/93\delta^{2}H_{p}$$

We conducted geographic assignment of feather δ^2^H_f_ values to estimated values of δ^2^H_p_ using the online workspace IsoMAP, which was publicly available at http://isomap.org. In the context of avian samples intercepted during the spring (vernal) season, our analysis was grounded in the ^2^H distribution map that corresponds to the model for the months of December and January. This model integrates a multitude of climate and geographical variables, encompassing precipitation trends, altitude, average ambient temperature, and geospatial coordinates expressed in terms of longitude and latitude. The selection of this model was predicated on the prevailing hypothesis suggesting that in Common Teal, the primary duration of winter molt, a significant physiological event in birds that results in changes in the ^2^H concentration in feathers, predominantly occurs within this temporal span [17,37].

Contrastingly, for bird specimens intercepted during the autumn (fall) season, we employed the analytical model corresponding to the months of June and July. This choice was based on the hypothesis that the majority of the summer molt, which involves shedding and replacement of plumage in birds, primarily occurs during these two months.

Considering that the sample sizes for both spring and autumn seasons were relatively small, we opted to analyze each sample separately. Leveraging the capabilities of QGIS, we individually visualized the probable origin of each bird. To identify the most probable region of origin for each specimen, we designated the top 10% of the spatial distribution of each sample as the most likely location of origin. The top 10% for each bird was assigned a value of 1, while the remaining 90% of the spatial distribution was assigned a value of 0. Following this, we compiled the probabilities for all samples, resulting in a cell grid that visually represents the potential regions of origin for the sample of birds. This procedure was carried out independently for both the spring and autumn seasons.

To augment the precision of the identified regions of origin, we incorporated additional data pertaining to the known habitat areas of the species, as well as specific ring recovery information from the study location. During our spatial analysis, we constructed a masking layer to obscure areas not recognized as typical habitats for Common Teal. The integration of ring recovery data allowed us to infer the most probable wintering locations for the species. As a culmination of our analysis, we demarcated several potential migratory routes (flyways) linking wintering regions to breeding areas.

### 2.5. Avian Influenza Virus Isolation

This study utilized the biosafety level-3 (BSL-3) facilities of the FRC FTM. Aliquots of each collected sample were used to isolate AIVs. For this purpose, samples were mixed using a vortex shaker and transferred to new 1.5 mL tubes following centrifugation for 3 min at 3000× *g*. Supernatants were transferred to a new 1.5 mL tube containing penicillin and gentamicin. SPF chicken embryos (3 per sample) were inoculated with 100 μL of sample in the allantoic cavity and incubated for 72 h in the BSL-3 laboratory of the FRC FTM [Terrestrial Code: Avian Influenza—WOAH—World Organisation for Animal Health, Available online: https://www.woah.org/en/disease/avian-influenza, accessed on 25 December 2023]. Allantoic fluid was collected in individual tubes and tested for hemagglutinating activity. After 3 serial passages of virus cultivation, all HA-positive samples were aliquoted for AIV M gene PCR testing.

### 2.6. Avian Influenza Virus Detection by RT-PCR and Sequencing

All samples with HA activity were tested for the presence of influenza A. For this, RNA was isolated from the allantoic fluid using the RIBO-sorb kit (AmpliSens, Moscow, Russia). The resulting RNA was used in the reverse transcription reaction using the REVERTA-L kit (AmpliSens, Russia). The presence of M gene regions of the influenza A virus was determined by real-time PCR using the AmpliSens Influenza virus A/B-FL kit (AmpliSens, Russia).

The genomes of the isolated strains were sequenced using next-generation sequencing, as previously reported by the National Institute of Animal Health, Tsukuba, Japan [43].

Nucleotide sequences of nine viruses characterized in this study have been deposited with the Global Initiative on Sharing All Influenza Data (GISAID) under the following ID: EPI_ISL_345915.

### 2.7. Genome Analysis and Phylogenetics of Avian Influenza Viruses

For each of the obtained genome segment sequences of the influenza viruses, the most identical sequences were identified using the BLAST algorithm (blastn program with an analysis depth of 50 items) [44] from the GISAID database [45]. The isolate from the database with the closest sequences and the closest correspondence to the following criteria were entered into the table for each segment of the genome for each sampled isolate: (1) the place of collection of the sample that best corresponds to the wintering of birds (breeding territories in Russia and Mongolia were excluded); and (2) the time of sample collection which is most appropriate for wintering birds (period November–March). Thus, the closest sequences related to the viruses from nesting areas in Siberia and Mongolia were excluded. For example, for the sample A/teal/Novosibirsk_region/816/2018_H3N8_MP, the closest identical sample A/common_teal/Novosibirsk_region/3324k/2020|A_/_H3N8||MP|2020-08-29|EPI_ISL_1184514 was excluded.

Next, after combining the files into a table using the SeqKit toolkit [46], repeated sequences by name were deleted.

The obtained sequence sets were subjected to multiple alignment using the MAFFT v7.520 multiple alignment algorithm (downloadable version) [47] using the default parameters: gap opening penalty, 1.53; gap extension penalty, 0.0. Then, files containing multiple alignments were opened in the Unipro UGENE program [48], where they were manually checked. After the final alignment, sites containing deletions were deleted.

Using the IQ-TREE web server [49], phylogenetic trees were constructed using the ML—maximum likelihood method—and branch support was evaluated using ultrafast bootstrap [50] and SH-aLRT branch test with the number of iterations equal to 1000. For each gene segment, the model of nucleotide substitutions was determined using ModelFinder via IQ-TREE [51]. The branches with values SH-aLRT > 80% and UFboot > 95% were considered relevant.

Tree visualization and topology analysis were performed using iTOL v.6 software [52].

Manipulations with files containing sequences were carried out, among other analyses, using Biopython 1.81 [53] and the SeqKit toolkit [46].

## 3. Results

### 3.1. Isotopic Composition of Feathers among Birds of Different Sex and Age

Among the birds tested, we analyzed the isotopic composition of the feathers of 18 samples from autumn migrants and 10 samples from spring migrants. There was no effect of age and sex on deuterium content in autumn samples. Also, no effect of sex was observed for vernal samples. No significant differences were found in δ^2^H_f_ in teals of different sexes, both in autumn and in spring (autumn: Mann–Whitney test, U = 1.33, n_1_ = 10, n_2_ = 8, *p* = 0.203; spring: U = 0.21, n_1_ = 4, n_2_ = 6, *p* = 0.914). Among the autumn teals, there were no differences in δ^2^H_f_ between yearlings and adults (Mann–Whitney test, U = 0.711; n_1_ = 10; n_2_ = 8; *p* = 0.515).

### 3.2. Deciphering the Values of the Deuterium Content in the Autumn Migrants’ Feathers

Analysis of the feathers of 18 autumn teals showed that the region of their most probable origin extends in a narrow strip from the south of Scandinavia to Tuva, covering the study region and approximately 500 km to the south (Figure 1).

The polygon presented in the figure is shaped in the latitudinal direction and does not include a significant part of the teal habitat located to the north and south of the sampling region. Also, all autumn samples depicted here were taken in a short period of 2–12 September, which is the initial stage of the fall migration of ducks in this region [28]. For the reasons and explanations see the Discussion Section.

### 3.3. Deciphering the Values of the Deuterium Content in the Spring Migrants’ Feathers

Analysis of 10 spring samples showed that examined individuals could winter in the vast area of southern Hindustan, the southern and western coasts of the Mediterranean, Indo-China, the coast of southeast Asia, and a few sites in northern Europe. These areas also include banding sites for wintering teals caught in the study region (Figure 2).

### 3.4. Detection of Avian Influenza Viruses in Common Teals

We took cloacal samples from 50 Common Teals which were obtained by hunters during the spring hunting season (April 2018) and from 73 Common Teals during the autumn hunting season (August–September 2018). We were unable to detect AIV in the spring samples, including those from which the feathers were collected. In nine autumn samples (12.3% of 73 inds.), we detected influenza virus RNA and isolated the virus on embryonated chicken eggs (Table 1). Next, we sequenced the complete genome of the isolates and determined their subtype, of which the most common subtype was H3N8 (7/9) (Table 1). The other two isolates had the subtypes H2N3 and H4N6. There was no effect of age and sex on AIV prevalence in autumn samples.

### 3.5. Genome Analysis and Phylogenetics of Avian Influenza Viruses

In order to identify the most likely phylogenetic relationship and the connection of the isolated viruses to the territories of origin and possible wintering sites of Common Teals, we conducted a BLAST analysis and indicated the highest percent of identical sequences (Table 2), according to the criteria and limitations described in the Methods Section, as well as a phylogenetic analysis. It must be said that in the BLAST analysis, we found that the most highly identical sequences to our isolates were strains isolated from wild ducks of different species in nesting areas in western Siberia and Mongolia. They were excluded from the analysis according to our criteria.

Almost half of the segments (34 out of 70) with highest percent of identical sequences in the table (97–99.7%) originated from India and Bangladesh, followed by segments originating from China and Korea (98–100%) with 6/72 and 8/72, respectively. A total of 14 out of 70 segments originated from European regions and northern Africa (Egypt). We found that virus 816 contained all the specific segments from isolates that were closest to the Asian wintering areas.

In contrast, virus 821 had the highest similarity to the European viruses, containing five of the eight segments closest to European viruses, with the exception of PB1 and MP. In addition, since the HA segment has the highest similarity to American strains, there is an assumed potential reassortment for this segment (marked with an asterisk in Table 2). Virus 821 had mostly segments found to have highest similarity with European regions, suggesting possible European origin and thus the westward migration of wild birds. 

The remaining six viruses had one or two European-originated segments, and the rest were Asian, indicating that most segments studied have close phylogenetic relationships with Asian variants. Out of all genome segments, the NP segment of studied viruses had the highest number of European variants according to the BLAST analysis (4 out of 9), followed by the pPB2 segment with 3 out of 9 and the rest of the segments with one out of 9.

In general, the mosaic pattern of the highest similarities of AIV segments to wintering areas indicates a greater percentage of Asian segment variants (ranged in 97–100%) in the genomes of viruses isolated from Common Teals in Siberia in autumn. All the segments more likely originated from wintering sites, according to the BLAST analysis that corresponds with our data determined by SIA and by ring recoveries according to Veen et al. [28]. Additionally, we evaluated the similarity between the HA segments for the most represented H3N8 virus (n = 7). It ranged between 98.4 and 100% (Appendix A). Strains 817 and 819 were found to be the most distant, and the group of strains comprising 823, 824, and 825 were the closest. Such results correlate with the topology of the tree (Figure 3).

We performed a phylogenetic analysis of all segments of the AIV genomes isolated from teals in Western Siberia (Figure 3, Appendix A).

In general, each analyzed segment on the tree was phylogenetically close to other viruses isolated from wild ducks in western Siberia (excluded from the BLAST analysis) and fall into the same clades that contain the viruses presented in Table 2. Thus, phylogenetic trees also show that most of the segments of the studied viruses isolated from teals have a close phylogenetic relationship with viruses isolated in the southern regions of wintering areas, in southeast Asian countries including India and Korea. Only a few segments are phylogenetically more closely related to viruses from European wintering areas.

## 4. Discussion

In this paper, we are the first to present some up-to-date data on the molting territories and, hence, the starting areas of migratory movements of the Common Teal in Asia. To our knowledge, this study is also the first to use stable isotope analysis to study the migration of ducks—avian influenza carriers. In addition, in our work, the migration routes of infected individuals were first analyzed simultaneously with those of uninfected individuals. The results obtained contribute to a better understanding of the migration of one of the main carriers of low pathogenicity avian influenza viruses.

We did not observe any differences between the place of origin of feathers either by sex or by the age of birds. Obviously, the size of the examined sample does not allow us to make any conclusions for the entire population. However, some differences in sex or age among autumn birds were quite expected, since adult males in Common Teals molt before females, and young birds migrate at the latest time [55]. Similar differences have been reported for many species of migratory birds [56]. Consequently, all individuals sampled in a relatively short (14 days) period in one place must have molted in different regions before. The observed absence of such differences could be explained either by the small sample size or the geographically close location of molting sites for all age and sex groups. In western Europe, European populations of Common Teals from different nesting areas can be found in the same locations during wintering or migrations [16]; among other duck species, both the mixing and segregation of local populations on wintering grounds are possible [57]. In our case, different sexual and age groups from one population possibly are mixing at the very beginning of fall migration. The lack of effect of molting area on the infestation of teals with LPAIV may be a result of small sample size. Also, ducks can be infected already during migration within one group originating from one molting area.

With regard to pre-breeding (winter) molting, it should be noted that the presence of the Mediterranean region or the coast of Southeast Asia in the area of high probability of molting for our samples is most likely to be an artefact caused by the patterns of the relative abundance of deuterium in coastal precipitations. This can be clarified by increasing the number of studied samples and the simultaneous use of GPS tracking to locate migration routes of Siberian Common Teals.

Nevertheless, this use of SIA, even on such a small sample, showed that Hindustan has been preserved for the past 40 years as the main wintering region for the Common Teal nesting in western Siberia. At the same time, within the subcontinent, some differences were found in the locations of wintering areas formerly reported by ring recovering. In particular, according to our stable isotope analysis, the places of completion of the pre-breeding molting of teals are restricted within the southern part of the peninsula, while the Indian ringing sites of ducks caught in our study region are mainly concentrated in Pench National Park (Central India, [28]). It is probable that the ducks there could have been massively trapped after they had already begun their spring migration. At the same time, these ring recoveries express nothing about the place of the pre-breeding molting of the trapped teals. The coastal localization of the pre-breeding molt of wintering teals suggests the possibility of their contact with local avifauna, including those migrating along the coast for long distances, such as tropical terns like Onychoprion [58], which may form the primary link in the transmission pathway for viral pathogens.

Virologic analysis showed that the segments of Common Teal viruses may have different origins. However, according to the BLAST analysis results, the majority of the segments have highest similarity (97–100%) to viruses from Asian regions and could be connected with southern and eastern directions of seasonal migrations to wintering sites. The limitation of such an approach is that some regions may be underrepresented in the database; however, comparative analysis will be more accurate with the more active accumulation of sequences in the database in the future.

This information may indirectly confirm the predominant role of teal migrations from western Siberia to Asian regions in autumn, as was found using SIA and ring recoveries. Such comparative phylogeographic studies of segments of the AIV host—Common Teal—could complement the results of stable isotope analysis and ring recoveries. However, it is necessary to develop a more accurate mathematical apparatus that reliably calculates and compares such data. The difficulty lies in the fact that the influenza virus has a segmented genome and has the property of reassortment, when segments can reassort between viruses of different origin in wintering areas (for example, originating from Europe or Asia).

Taken together, our data give us an attractive opportunity to consider the relatively high probability density of pre-breeding molting in southeast China as evidence of direct migrations of the Common Teal from the nesting sites in western Siberia to southeast China. However, there are no ring recoveries to support this, and the results of the satellite tracking of teals tagged with GPS trackers on Lake Poyang do not confirm their flights even to Lake Qinghai (that is, there is no evidence of even a first step toward the Dzungarian Gates to reach Siberia, [23]). In the absence of direct confirmation, these SIA results tend to be an artefact of the high variability in the isotope landscape of precipitation in Asia [27]. Apparently, the assumption about the direct migration of west Siberian Anatidae through the Dzungarian Gates to Southeast Asia requires more detailed verification with the simultaneous use of SIA feathers and the distant tracking of birds trapped directly on the Dzungarian migration route.

The significant latitudinal extent of the polygon with the highest probability density of post-breeding molting is most likely due to the high variability in the isotope landscape in Asia, as well as to the small sample size presented here. Nevertheless, the ringing recoveries data [27] suggest the origination of the samples outside of the study area as unlikely. In addition, according to [17], Common Teals that molt and nest in the north of the European part of Russia migrate for wintering mostly westward and not to Siberia. Thus, we can constrain the final area of the samples’ origination in the longitudinal direction by the actual region of research. And the localization of the obtained polygon within western Siberia allows us to exclude the migration of the examined individuals from the forest or forest–tundra zone. This looks equivocal compared with the observation that the breeding area of the Common Teal in Siberia stretches to the lower reaches of the Ob river, as well as with the fact that Teals migrate from the study area to the north and east, according to ring recoveries [28]. However, all our autumn teals were sampled from September 2 to 12. In the study region, this is the very beginning of the autumn migration of the species. It is possible that more northern subpopulations migrate in a later period, similar to the Common Pochard from the European part of Russia [57]. However, this suggestion, as well as the others mentioned above, also needs to be further tested on a larger sample size. Undoubtedly, we should keep in mind the limitations of the study (such as sample size, number and identity of sequences, and limited territory), because it is a pilot study trying to utilize an alternative approach for the investigation of host and pathogen ecology that has significance for future research.

## 5. Conclusions

Despite the small size of the sample, this study clarified some crucial issues concerning the locations of pre- and post-breeding molting of the Common Teal. It seems that autumn-hunted prey in the study area generally consist of local nesters instead of northern migrants. For spring migrants, throughout the study area, the most probable wintering grounds are the southern coasts of Hindustan, southeast Asia, and the Mediterranean. The occurrence of wintering in the last two regions requires confirmation. The wintering of west Siberian teals on the coast of Hindustan suggests the possibility that pathogens circulating along the coast within local avifauna are transmitted to west Siberia. The wintering of the same population of teals in southeast China, if confirmed, means direct contact with local carriers in the pesthole of highly pathogenic strains like H5N1. The results of the phylogenetic analysis of the AIV genome segments isolated from Common Teals in western Siberia correlates with the data from the stable isotope analysis. It seems that the additional evidence of the connection of teals with southeast Asia can be considered:A very high level of nucleotide sequence identity (97–100%) for a certain AIV segment (direct connection).The presence of a strain with all segments from the southeast Asian region. A further detailed study on the flight paths of the Common Teal from western Siberia to other regions of Eurasia with the use of large samples and an enlarged network of sampling points is required.

## Figures and Tables

**Figure 1 microorganisms-12-00357-f001:**
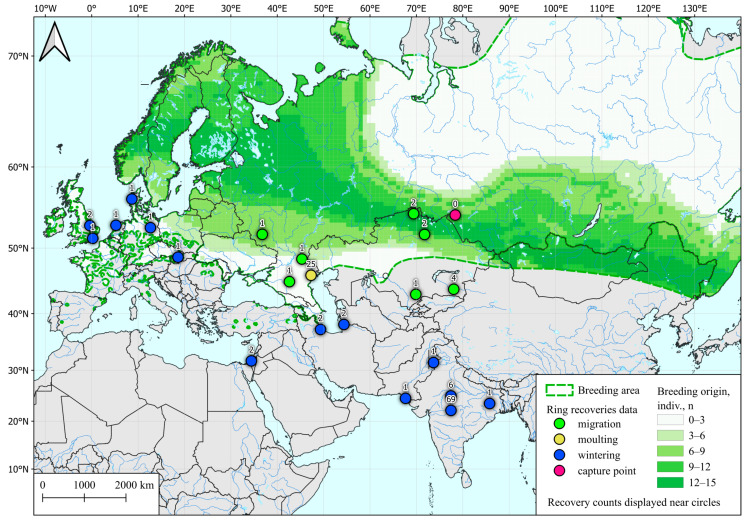
Possible post-breeding molting areas for Common Teal (*Anas crecca*) captured in western Siberia during fall migration, determined by SIA and by ring recoveries according to Veen et al. [28]. Common Teal area data are taken from [54].

**Figure 2 microorganisms-12-00357-f002:**
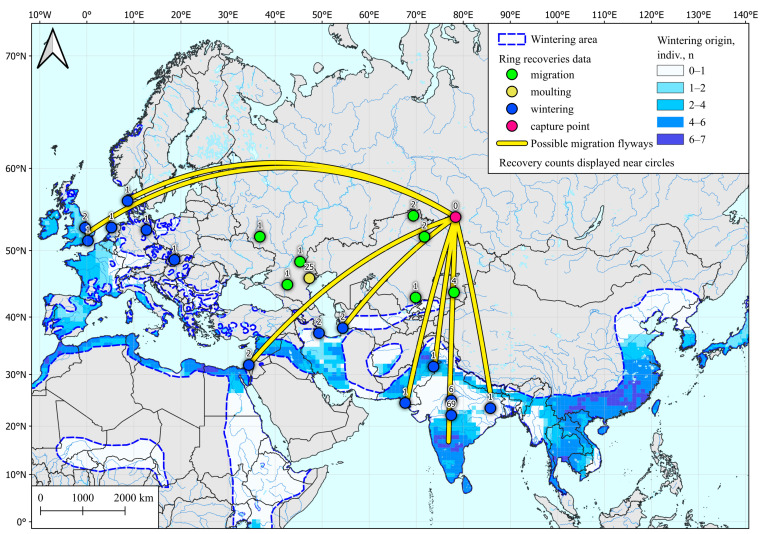
Wintering area for Common Teal (Anas crecca) captured in spring in Western Siberia, determined by SIA and by ring recoveries according to Veen et al. [28]. Common Teal area data are taken from [54].

**Figure 3 microorganisms-12-00357-f003:**
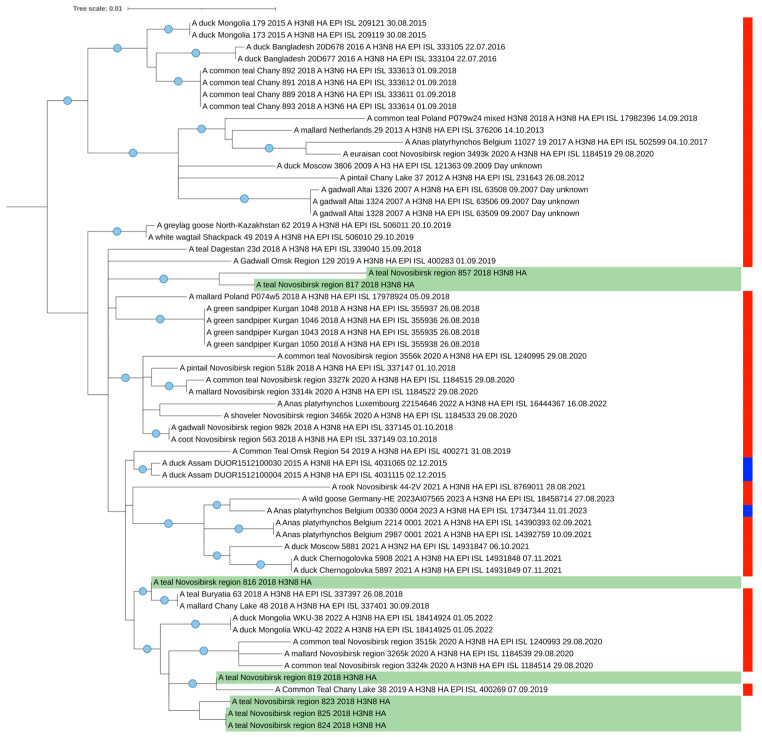
Maximum likelihood phylogenetic tree of HA segment of seven H3N8 AIVs isolated from Common Teals in western Siberia (marked green). The blue circle symbol indicates branches with values SH-aLRT > 80% and UFboot > 95%. The color markings on the right indicate the time of sample collection (red is the summer nesting period, blue is the winter wintering period).

**Table 1 microorganisms-12-00357-t001:** Avian influenza viruses isolated from Common Teals (*Anas crecca*) in autumn 2018 in western Siberia.

Virus	Abbreviation	Subtype	Accession Number	Sex of Host	Age Host
A/teal/Novosibirsk_region/715/2018 (H4N6)	715	H4N6	EPI_ISL_18799160	male	juv
A/teal/Novosibirsk_region/816/2018 (H3N8)	816	H3N8	EPI_ISL_18794362	male	juv
A/teal/Novosibirsk_region/817/2018 (H3N8)	817	H3N8	EPI_ISL_18794363	female	ad
A/teal/Novosibirsk_region/819/2018 (H3N8)	819	H3N8	EPI_ISL_18794364	male	ad
A/teal/Novosibirsk_region/821/2018 (H2N3)	821	H2N3	EPI_ISL_18794365	male	ad
A/teal/Novosibirsk_region/823/2018 (H3N8)	823	H3N8	EPI_ISL_18794366	female	ad
A/teal/Novosibirsk_region/824/2018 (H3N8)	824	H3N8	EPI_ISL_18794367	female	ad
A/teal/Novosibirsk_region/825/2018 (H3N8)	825	H3N8	EPI_ISL_18794368	female	juv
A/teal/Novosibirsk_region/857/2018 (H3N8)	857	H3N8	EPI_ISL_18794370	male	juv

**Table 2 microorganisms-12-00357-t002:** Results of BLAST analysis showing nucleotide sequences that were the most identical to the segments of studied avian influenza viruses isolated from Common Teals (*Anas crecca*) in autumn 2018 in western Siberia. GISAID accession numbers (indicated without EPI_ISL_ prefix), % of identical sequences, and geographical area of isolation are presented. Cells containing samples from the Asian part of the continent, as well as one sample from the Australian region, are colored grey. The cells containing samples from the European part of the continent, as well as one sample from North America, are colored white.

Virus	PB2	PB1	PA	HA	NP	NA	MP	NS
715	Bangladesh 333,102 98.901%	Russia, Dagestan 331,290 98.509%	Bangladesh 503,320 97.309%	Bulgaria 174,868 93.143%	Bangladesh 503,481 99.595%	Korea 309,224 98.200%	China, Shandong 12,627,550 99.230%	Korea 14,835,465 100.000%
816	Bangladesh 387,988 99.560%	Bangladesh 503,527 99.235%	Bangladesh 503,510 98.391%	India, Assam 4,031,065 99.595%	n.d.	China, Guangxi 273,054 98.017%	Bangladesh 503,538 99.483%	Korea 14,835,573 99.876%
817	Netherlands 267,243 97.723%	Bangladesh 4,071,629 99.337%	Bangladesh 503,499 97.982%	India, Assam 4,031,065 98.817%	China, Hubei 363,770 98.247%	China, Guangxi 273,054 98.017%	Japan, Tottori 503,006 99.389%	Bangladesh 9,953,028 99.502%
819	Russia, Dagestan 331,290 99.313%	Bangladesh 333,102 99.096%	King Island 14,768,770 97.749%	India, Assam 4,031,115 99.158%	Japan, Tottori 503,006 99.326%	India, Assam 4,031,065 99.137%	Japan, Hokkaido 16,146 99.183%	Bangladesh 503,538 99.354%
821	Belgium 502,606 98.304%	China, Shandong 12,627,550 99.074%	Belgium 502,608 97.596%	USA, South Carolina 397,752 97.849%	Germany 18,458,713 98.653%	Ukraine 4,056,412 99.196%	Korea 14,835,569 99.070%	Netherlands 243,393 99.625%
823	Bangladesh 387,988 99.371%	Bangladesh 503,509 98.911%	India, Assam 4,031,115 97.884%	India, Assam 4,031,115 99.348%	Egypt 387,968 98.597%	n.d.	Bangladesh 387,988 99.695%	Korea 14,835,462 100.000%
824	Bangladesh 387,988 99.339%	Bangladesh 503,509 98.911%	India, Assam 4,031,115 97.934%	India, Assam 4,031,065 99.354%	Egypt 387,968 98.664%	India, Assam 4,031,065 99.059%	Bangladesh 387,988 99.387%	Korea 14,835,462 100.000%
825	Bangladesh 387,988 99.339%	Bangladesh 503,509 98.911%	India, Assam 4,031,115 98.024%	India, Assam 4,031,115 99.398%	Egypt 387,968 98.653%	India, Assam 4,031,065 99.059%	Bangladesh 387,988 99.686%	Korea 14,835,462 100.000%
857	Egypt 17,508,501 99.378%	Bangladesh 503,526 98.632%	Bangladesh 503,525 97.419%	India, Assam 4,031,065 98.807%	China, Hubei 363,770 98.318%	India, Assam 4,031,065 99.140%	Dagestan 331,307 99.047%	Korea 4,071,213 99.499%

n.d.—not determined, due to low quality of sequence data for analysis.

## Data Availability

All sequences from the study are available in GISAID (accession numbers: EPI_ISL_345915-EPI_ISL_345915).

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
