# Peer review of "Stable Isotope Analysis Reveals Common Teal (Anas crecca) Molting Sites in Western Siberia: Implications for Avian Influenza Virus Spread"

_microorganisms, 2024, doi:10.3390/microorganisms12020357_

Round 1
Reviewer 1 Report
Comments and Suggestions for Authors
General Comments
The manuscript (ms) entitled "Mapping molting sites of Eurasian Teal (Anas crecca) in Western Siberia by stable isotope analysis: application to study of the avian influenza virus spread" (MS ID: microorganisms-2819530) is quite interesting for the emerging field of low pathogenic avian influenza viruses (LPAIV) detections and relevant respond to possible waterfowl outbreaks by understanding of the migration of Anas crecca, one of the main carriers of LPAIV. As far as I know, it is the first study that actually presents the use of stable isotope analysis to study the migration pattern of ducks-avian influenza carriers and compare migration routes of infected and uninfected individuals of a species.
In general, ms is well structured and refer to other relevant studies with few exceptions. Data set as well as methodology used are adequate to sustain discussion and conclusions. I also have some minor specific comments as you can check below:
Specific Comments
Title: I suggest to use “Common Teal” as common name of the target species. As you can check through Keller et al (2020), the species is now considered distinct from the North American Green-winged Teal (Anas carolinensis). I also suggest to replace “Eurasean Teal” with “Common Teal” in the whole ms so as to avoid misunderstandings for the readers.
Keller, V., Herrando, S., VoÅ™íšek, P., Franch, M., Kipson, M., Milanesi, P., Martí, D., Anton, M., Klvanova, A., Kalyakin, M., Bauer, H., & Foppen, R. (2020). European Breeding Bird Atlas 2: Distribution, Abundance and Change.
Introduction
Page 3, Lines 95-98. “Mapping of ring recoveries … and even Northern Europe” Please add relevant reference(s).
Page 3, Line 100: Give latin-scientific names for “Common Pochard, Common Pintail, Wigeon”
Page 3, Line 115: Give latin-scientific name for “Bean Goose”
Materials and Methods
Page 2, Lines 158-160. “For the isotope analysis … totally 10 spring samples”. Please give more details about the Primary feather parts cut off. For example, how much in weight and length was removed per feather and which part?
Results
Page 8, Lines 296-298. Clarify to the legend of the Figure 1 the meaning of the numbers of the spots. Give also an orientation symbol on the map.
Page 9, Lines 307-308. Clarify to the legend of the Figure 2 the meaning of the numbers of the spots. Give also an orientation symbol on the map.
Page 9, Lines 331-333. Clarify to the legend of Table 2 the meaning of the different colour of the cells
Page 9, Line 336. What do you mean? Please clarify.
Author Response
Specific Comments
Title: I suggest to use “Common Teal” as common name of the target species. As you can check through Keller et al (2020), the species is now considered distinct from the North American Green-winged Teal (Anas carolinensis). I also suggest to replace “Eurasean Teal” with “Common Teal” in the whole ms so as to avoid misunderstandings for the readers.
Keller, V., Herrando, S., VoÅ™íšek, P., Franch, M., Kipson, M., Milanesi, P., Martí, D., Anton, M., Klvanova, A., Kalyakin, M., Bauer, H., & Foppen, R. (2020). European Breeding Bird Atlas 2: Distribution, Abundance and Change.
Response: thank you for the comment. We compared different publications and agreed that name “Common Teal” is more generally used in scientific literature. The name “Eurasean Teal” is replaced with “Common Teal” in the whole ms
Introduction
Page 3, Lines 95-98. “Mapping of ring recoveries … and even Northern Europe” Please add relevant reference(s).
Page 3, Line 100: Give latin-scientific names for “Common Pochard, Common Pintail, Wigeon”
Page 3, Line 115: Give latin-scientific name for “Bean Goose”
Response: thank you for the comment. We agree and tried to modify the text and responded to several similar comments above (please see)
Materials and Methods
Page 2, Lines 158-160. “For the isotope analysis … totally 10 spring samples”. Please give more details about the Primary feather parts cut off. For example, how much in weight and length was removed per feather and which part?
Response: text was corrected, required information was added to p. 2.2. in Methods
Results
Page 8, Lines 296-298. Clarify to the legend of the Figure 1 the meaning of the numbers of the spots. Give also an orientation symbol on the map.
Page 9, Lines 307-308. Clarify to the legend of the Figure 2 the meaning of the numbers of the spots. Give also an orientation symbol on the map.
Response: Corrected according to the comment (please see the legends in Fig.1 and Fig.2)
Page 9, Lines 331-333. Clarify to the legend of Table 2 the meaning of the different colour of the cells
Response: Corrected according to the comment (please see the legend)
Page 9, Line 336. What do you mean? Please clarify.
Response: Corrected according to the comment (please see the legend)
Reviewer 2 Report
Comments and Suggestions for Authors
The authors mapped the molting sites of Eurasian Teal by isotope analysis and compared these results with the genetic diversity of Influenza virus found in the birds. This study is at the interception of animal behavior and virology. Some concerns should be addressed before publication.
1. Page 1, affiliations: please use the same font throughout the list.
2. Page 1: the corresponding author is not mentioned.
3. Some references are cited in red.
4. Page 6: why the authors performed viral isolation on embryonated eggs instead of directly real time PCR? The protocol of culture is more cumbersome and probably less sensitive than the real time PCR one.
5. Page 6, line 251: only one accession number is provided.
6. Page 8, Figure 1: the resolution of the figure is not optimal. Also, the circles of migration in green are difficult to detect on the green background.
7. Page 9, Table 2. The authors should eliminate the EPI_ISL_ prefix of each accession number, include it in the legend and add in each case only the number, for a better comprehension. That would allow to include the percent identity for each Blast analysis. The colors (blue and red) of the table should be explained.
8. One limitation of the Blast analysis is that isolates from some regions may be underrepresented in the GISAID database. Do the authors that this may have occurred?
9. Page 10, line 336: this line is out of context or it refers to Table 2, apparently to two segments that could not be analyzed, probably because of not enough coverage of the sequence?
10. Page 12, Figure 3: The legend of the figure should include the model used for the phylogenetic inference.
11. Page 12, Figure 3: the legend mention that 7 genomes are shown, the 7 H3N8 HA genomes, but only one is shown. What was the percent identity between the 7 segments? The figure contains also blue circles and red and blue lines, but none of these colors are explained in the legend.
12. The nucleotide identity of the segments with their related sequences found in GISAID is not mentioned in the Table, nor in Results nor in Conclusions.
13. Did the authors performed also phylogenetic analysis for the 2 other isolates belonging to other subtypes (H4N6 and H2N3)?
Author Response
- Page 1, affiliations: please use the same font throughout the list.
Response: corrected
- Page 1: the corresponding author is not mentioned.
Response: corrected
- Some references are cited in red.
Response: corrected
- Page 6: why the authors performed viral isolation on embryonated eggs instead of directly real time PCR? The protocol of culture is more cumbersome and probably less sensitive than the real time PCR one.
Response: There are numerous studies comparing the virus isolation method using chicken embryos and PCR. A number of works indicate a greater sensitivity of PCR diagnostics (for example, Kim et al., 2019, doi: 10.4142/jvs.2019.20.e56), others indicate a higher sensitivity of isolation in chicken embryos (for example, Spackman et al., 2003, DOI: 10.1637/0005-2086-47.s3.1079). On average, the results show that they can be compared in sensitivity. Obviously, separate methods of detection have some limitations. Therefore, different methods should be combined for optimal surveillance. We choose VI method with further PCR and sequencing because tried not only detect but also to isolate the virus so that we could then have sufficient concentration and sequence the complete genome. It is much more difficult to sequence the complete genome from the original specimens.
- Page 6, line 251: only one accession number is provided.
Response: all accession numbers are provided
- Page 8, Figure 1: the resolution of the figure is not optimal. Also, the circles of migration in green are difficult to detect on the green background.
Response: We thank you for your comment. Indeed, the resolution turned out to be lower than required, this has been corrected. As for the green circles, their correspondence with the green background of the polygons of the probable origin of the feathers was chosen just to emphasize the proximity of the processes of molting and the first steps of autumn migration. At the same time, shades of green make it possible to confidently distinguish circles even against the backdrop of a polygon. Therefore, we decided not to change the color palette of the picture.
- Page 9, Table 2. The authors should eliminate the EPI_ISL_ prefix of each accession number, include it in the legend and add in each case only the number, for a better comprehension. That would allow to include the percent identity for each Blast analysis. The colors (blue and red) of the table should be explained.
Response: Thank you for fine suggestion! We removed the EPI_ISL_ prefix and added identity percent. Additionally we changed the colors and added the explanation in Legend. The color has been changed with the aim not to confuse the reader, since red and blue are used in the Figures to indicate the season of isolates in phylogenetic trees (please see Figure 3 for example).
- One limitation of the Blast analysis is that isolates from some regions may be underrepresented in the GISAID database. Do the authors that this may have occurred?
Response: Indeed, some regions may be underrepresented in the database. We have added this limitation to the text of the article (Discussion section). However, we believe that this approach will be useful in the future, when more and more sequences will appear in the database and comparative analysis will be more accurate with the accumulation of sequences in the database. We added: “The limitation of such approach is that some regions may be underrepresented in the Database, however comparative analysis will be more accurate with the more active accumulation of sequences in the database in future.”
- Page 10, line 336: this line is out of context or it refers to Table 2, apparently to two segments that could not be analyzed, probably because of not enough coverage of the sequence?
Response: we agree, we corrected the text. Indeed, several segments were not included in analysis due to not enough quality of sequence data.
- Page 12, Figure 3: The legend of the figure should include the model used for the phylogenetic inference.
Response: we added the details in the Legend according to the comment
- Page 12, Figure 3: the legend mention that 7 genomes are shown, the 7 H3N8 HA genomes, but only one is shown. What was the percent identity between the 7 segments? The figure contains also blue circles and red and blue lines, but none of these colors are explained in the legend.
Response: Indeed, this is our mistake, we mistakenly placed the NA tree of the N3 subtype, instead of the HA segment tree from seven H3N8 strains. Now we have correctly put the HA tree required and added all the required information to the caption of Figures.
According to the comment on percent identity between the 7 HA segments (H3N8) – we added matrices of pairwise distances between HA segments of Western Siberian H3N8. Additionally, we evaluated the identity between the HA segments for the most represented H3N8 virus subtypes (h=7). It was ranged of 98.4%-100%. Strains 817 and 819 were found to be the most distant, and the group of strains 823, 824, 825 were the closest. Such results correlate with the topology of the tree (Figure 3)
- The nucleotide identity of the segments with their related sequences found in GISAID is not mentioned in the Table, nor in Results nor in Conclusions.
Response: We added identity in the Table. We corrected this flaw and provided a description in the table and results, and added detail to the conclusion. The highest identity of AIV segments in wintering areas indicates a greater percentage of Asian segment variants (ranged in 97%-100%) in the genomes of viruses isolated from Asian region.
- Did the authors performed also phylogenetic analysis for the 2 other isolates belonging to other subtypes (H4N6 and H2N3)?
Response: Sure, the results provided in Supplementary Materials (Figures S2-S5)
Reviewer 3 Report
Comments and Suggestions for Authors
The manuscript addressed
- Stable Isotope Tracing of Eurasian Teal Molting Sites in Western Siberia: A Tool for Avian Influenza Spread Assessment.
- The following points need to be considered by the authors
The title should be more informative like (Stable Isotope Analysis Reveals Eurasian Teal Molting Sites in Western Siberia: Implications for Avian Influenza Virus Transmission )
Abstract:
Please consider incorporating maps highlighting the identified molting sites, migration routes, and viral connections.
Some sentences could be shortened for improved readability, such as merging sentences 25 and 26 (e.g., "Post-breeding molt... occurred within the study region, potentially linking to their wintering areas in Southeast Asia, as supported by viral analyses.").
Introduction:
Please briefly mention a specific zoonotic disease relevant to your research could further engage the reader.
Consider ending the introduction with a thought-provoking question or highlighting a specific gap in knowledge your research addresses.
Please expand the drivers of zoonotic transmission, even in a single sentence, could provide more context and highlight the urgency of the research.
Results:
- The sample size for spring migrants (n=4) seems quite small to draw definitive conclusions.
- Please state the p-values.
- Briefly describe the key features of the map in Figure 1, highlighting how it supports the conclusion about the most probable origin region.
- Please discuss how the identified wintering areas in Southeast Asia align with the research objective of understanding potential AIV transmission pathways. This would connect the results to the broader context of zoonotic disease control.
- Figure 3 needs to be more clear.
- Discussion
- Please provide a significance of the use of stable isotope analysis, a powerful tool for tracing migratory movements and identifying potential virus reservoirs.
- The authors said (Obviously, the size of the examined sample does not allow us to make any conclusion for the entire population) I wonder why a larger samples didn't used in this study?
Comments on the Quality of English Language
Moderate changes are needed.
Author Response
The title should be more informative like (Stable Isotope Analysis Reveals Eurasian Teal Molting Sites in Western Siberia: Implications for Avian Influenza Virus Transmission )
Response: we modified the title according to the comment
Abstract:
Please consider incorporating maps highlighting the identified molting sites, migration routes, and viral connections.
Response: We have added a graphical abstract reflecting the main result of the article, in particular, the overlapping of the Common Teal migration map and the points of the isolation of several LPAIV strains
Some sentences could be shortened for improved readability, such as merging sentences 25 and 26 (e.g., "Post-breeding molt... occurred within the study region, potentially linking to their wintering areas in Southeast Asia, as supported by viral analyses.").
Response: We agree, abstract was corrected in a recommended way
Introduction:
Please briefly mention a specific zoonotic disease relevant to your research could further engage the reader.
Response: we added the example
Consider ending the introduction with a thought-provoking question or highlighting a specific gap in knowledge your research addresses.
Response: thank you so much! We have changed the final sentence according to the suggestion
Please expand the drivers of zoonotic transmission, even in a single sentence, could provide more context and highlight the urgency of the research.
Response: we added the sentence, providing details to highlight the urgency of the research.
Results:
The sample size for spring migrants (n=4) seems quite small to draw definitive conclusions.
Response: we added the text according to the comment. And also according to the comment of Reviewer 1 about limitation of sequences we agree. Despite the small number of samples of spring migrants, we made such an analysis as an example of a study that can be developed. In the future, when analyzing a large number of virus samples and sequences, the results will be more accurate.
Therefore, we have indicated this limitation of the study at this stage and its significance for future research in the text (see text).
Please state the p-values.
Response: We added description of statistical treatment to Methods, where also stated p-values threshold.
Briefly describe the key features of the map in Figure 1, highlighting how it supports the conclusion about the most probable origin region.
Response: These reasons are completely highlighted in Discussion, lines 818-831. Also we summarized the kee features of the Figure in Results
Please discuss how the identified wintering areas in Southeast Asia align with the research objective of understanding potential AIV transmission pathways. This would connect the results to the broader context of zoonotic disease control
Response: Since we identified the highest percentage of identity of viral segments from teal, isolated in Western Siberia with strains from the Asian region (e.g. India and China), this correlates with the data obtained by isotope analysis, which shows the most likely main wintering grounds in the Asian region. We believe that this indirectly confirms the circulation and transmission of viruses between these territories. This is discussed in original text in lines 801-812
Figure 3 needs to be more clear.
Response: we added the details in the Legend according to the comment, please see comment of Reviewer 1 above
Discussion
Please provide a significance of the use of stable isotope analysis, a powerful tool for tracing migratory movements and identifying potential virus reservoirs.
Response: The significance is that by using stable isotope analysis we can further confirm the overwintering data obtained through previous migration studies (based on ring recoveries) and link the waterfowl migration rules with the most likely sources of viruses from the wintering areas.
The authors said (Obviously, the size of the examined sample does not allow us to make any conclusion for the entire population) I wonder why a larger samples didn't used in this study?
Response: we completely agree with the comment. The number of samples is a limitation. This is very important for the accuracy of the study. The same comment as above. This is a pilot study, so we point to a methodological possibility based on a limited number of samples. In the future, it will be expanded and clearer conclusions and results will be obtained.
Reviewer 4 Report
Comments and Suggestions for Authors
The study of Druzyaka et al., titled “Mapping molting sites of Eurasian Teal (Anas crecca) in Southwestern Siberia (SWS) by stable isotope analysis: application to study of the avian influenza virus (AIV) spread” aimed to map the molting sites of migrating Eurasian Teals in the region by analyzing stable hydrogen isotope content in feathers and examining the genetic structure of viruses isolated from teals. The findings indicate that the probable pre-breeding molting grounds of spring Teals are likely located south of Hindustan, with post-breeding molt occurring within the study region. Phylogenetic analysis of AIV reveals a close relationship between SWS isolates and viruses from South and Southeast Asia, aligning with stable isotope mapping data. Most viral segments show high genetic identity and close phylogenetic relationships with viruses from Teal wintering areas in Southeast Asian countries, India, and Korea. The study suggests that the winter molt of SWS breeders on the Hindustan coast may involve interactions with local avifauna, potentially serving as a vector for AIV transmission within Eurasia. This study is important and can be accepted after minor revisions.
Minor revisions
1) Why didn't the authors add the term “Southwestern Siberia (SWS)” to the title of the manuscript instead of “Western Siberia”?
2) The limitation of analyzing only two regions and two periods may have influenced the results. Novosibirsk (53.60 N and 77.60 E, sampling area approximately 155 x 50 km, in autumn 2017 and spring and autumn 2018) and Tomsk region (57.60 N and 83.90 E, collection area about 15 x 20 km, only in spring 2018). Sampling occurred from September 2 to 12, 2017, and from April 20 to 30, 2018.
3) The number of samples collected is also small: For isotope analysis, 7th and 8th primary flight feathers were collected from each autumn migrant, totaling 18 samples; for spring migrants, we selected plumage parts, which tend to be the last to grow during pre-breeding molting, totaling 10 spring samples. During avian influenza monitoring in 2018, 50 Eurasian Teals were sampled in the spring season, and 73 Eurasian Teals preyed upon by amateur hunters were collected. This could pose challenges in differentiating species, sex, and age of the animals. The authors are encouraged to discuss the potential implications of this limitation and propose ways to address it.
4) The number of genomes obtained and analyzed in the study is also small. The authors are encouraged to acknowledge this limitation and discuss its potential impact on the robustness of their conclusions.
5) The study uses stable isotope analysis to determine bird migration and correlate it with the viruses detected in the study and GenBank data. It is positive to use host migration data to determine viral spread. However, this is usually done through phylogenetic analyses of viral spread and molecular clock analyses. The authors are encouraged to provide information on why they did not employ these approaches and to consider comparing them with the proposed new methodology, i.e., stable isotope analysis and viral genetic data, discussing the strengths and limitations of each approach.
Author Response
1) Why didn't the authors add the term “Southwestern Siberia (SWS)” to the title of the manuscript instead of “Western Siberia”?
Response: The molting sites are wider than the South, so we ask to keep the “Western Siberia”
2) The limitation of analyzing only two regions and two periods may have influenced the results. Novosibirsk (53.60 N and 77.60 E, sampling area approximately 155 x 50 km, in autumn 2017 and spring and autumn 2018) and Tomsk region (57.60 N and 83.90 E, collection area about 15 x 20 km, only in spring 2018). Sampling occurred from September 2 to 12, 2017, and from April 20 to 30, 2018.
Response: thank you for the comment. We agree and tried to modify the text and responded to several similar comments above (please see)
3) The number of samples collected is also small: For isotope analysis, 7th and 8th primary flight feathers were collected from each autumn migrant, totaling 18 samples; for spring migrants, we selected plumage parts, which tend to be the last to grow during pre-breeding molting, totaling 10 spring samples. During avian influenza monitoring in 2018, 50 Eurasian Teals were sampled in the spring season, and 73 Eurasian Teals preyed upon by amateur hunters were collected. This could pose challenges in differentiating species, sex, and age of the animals. The authors are encouraged to discuss the potential implications of this limitation and propose ways to address it.
Response: thank you for the comment. We agree and tried to modify the text and responded to several similar comments above (please see)
4) The number of genomes obtained and analyzed in the study is also small. The authors are encouraged to acknowledge this limitation and discuss its potential impact on the robustness of their conclusions.
Response: thank you for the comment. We agree and tried to modify the text and responded to several similar comments above (please see)
5) The study uses stable isotope analysis to determine bird migration and correlate it with the viruses detected in the study and GenBank data. It is positive to use host migration data to determine viral spread. However, this is usually done through phylogenetic analyses of viral spread and molecular clock analyses. The authors are encouraged to provide information on why they did not employ these approaches and to consider comparing them with the proposed new methodology, i.e., stable isotope analysis and viral genetic data, discussing the strengths and limitations of each approach.
Response: This fine comment summarizes the study, its limitations actually. We are grateful to Reviewer for this. We completely agree and tried to modify the text and responded to several similar comments above (please see). We believe this is interesting for readers, because it is some pilot study trying to have alternative approach for investigation of host and pathogen ecology. In the future, it could be expanded and clearer conclusions and results will be obtained to add the picture.
Round 2
Reviewer 2 Report
Comments and Suggestions for Authors
The authors addressed satisfactorely the comments.
Reviewer 3 Report
Comments and Suggestions for Authors
The manuscript is significantly improved to the satisfaction of the reviewer.